# Longitudinal Profiles of Recovery-Enhancing Processes: Job-Related Antecedents and Well-Being Outcomes

**DOI:** 10.3390/ijerph20075382

**Published:** 2023-04-03

**Authors:** Ulla Kinnunen, Anne Mäkikangas

**Affiliations:** 1Faculty of Social Sciences, Psychology, Tampere University, 33014 Tampere, Finland; 2Faculty of Social Sciences, Work Research Centre, Tampere University, 33014 Tampere, Finland

**Keywords:** detachment, exhaustion, LPA, physical activity, recovery, sleep, vigor, vitality

## Abstract

The present study aimed to examine longitudinal recovery profiles based on three recovery-enhancing processes, i.e., psychological detachment from work, physical exercise, and sleep. In addition, we examined whether job-related demands and resources predict profile membership and whether profile membership predicts well-being outcomes. The participants were Finnish employees (N = 664) who filled in an electronic questionnaire in three successive years. Latent profile analysis (LPA) revealed five stable profiles of recovery-enhancing processes across time: (1) physically inactive, highly detaching (15%), (2) impaired recovery processes (19%), (3) enhanced recovery processes (25%), (4) physically active, poorly detaching and sleeping (19%), and (5) physically active (29%). In addition, job-related antecedents and well-being outcomes showed unique differences between the five profiles identified. Altogether, our study takes recovery research a step forward in helping to understand how recovery-enhancing processes function simultaneously over the long-term and suggests that, from the perspective of well-being, detachment from work and good sleep are more crucial recovery processes than physical activity.

## 1. Introduction

Employees have to put their energy and other resources in use to perform effectively at work [1]. This resource expenditure is manifested in short-term strain reactions, of which employees typically recover after work. Thus, the recovery process is opposite to the strain process [2], during which short-term strain reactions are mitigated and they likely do not result in long-term strain reactions [3]. However, in today’s working life, there are several factors threatening successful recovery. Due to blurring boundaries between work and private life, work is extending its effects on private life, decreasing quantity and impairing quality of leisure time. Working life is also hectic, demanding, and under constant change, for example, due to rapid digitalization. All these factors impose strain on employees. Therefore, the question of how to recover efficiently from work strain has become crucial.

Recovery-enhancing processes can help recovery from work strain [3,4]. On the basis of a thorough review, Sonnentag [3] concluded that the most important of these recovery-enhancing processes are psychological detachment from work, physical activity, and sleep. Thus far, various recovery-enhancing processes have been mostly studied in isolation, although recovery processes are not separate from each other. For example, physical activity during leisure time helps to detach oneself from work [5,6] and to sleep better; see [7] for a meta-analysis. In addition, the processes may interact in explaining well-being outcomes, as occurred in the study by Cho and Park [8], in which physical exercise, together with either psychological detachment from work or good sleep during the weekend, was associated with a decrease in negative affect. Thus, several simultaneous recovery-enhancing processes may be needed to attain a positive well-being outcome.

The aim of the present study is to address this gap in the recovery research, that is, we examine recovery profiles based on psychological detachment from work, physical exercise, and sleep. This kind of person-centered research, of which the aim is to reveal which kind of combinations of constructs exist within individuals [9,10], is significant. For example, in our case, people have likely several types of recovery-enhancing processes concurrently in use, and certain combinations of processes may be more beneficial than others. Thus, person-centered analyses provide a uniquely informative perspective to recovery research by focusing on variable configurations. Besides identifying the longitudinal profiles (across two years) of psychological detachment from work, physical exercise, and sleep, we also study antecedents (job demands and resources) and well-being outcomes (job exhaustion, vigor at work, and vitality) of the profiles. In doing so, we aim to validate the profiles in relation to theoretically relevant antecedents and outcomes. In summary, our study contributes to the existing recovery literature in three ways. First, our study reveals the various combinations of recovery-enhancing processes among individuals. Second, we indicate how job demands and resources function as antecedents of profile membership, and, third, how profile membership predicts well-being outcomes. This kind of knowledge is crucial from the viewpoints of recovery theory and practice. In addition, our study contributes to the methodology: it utilizes longitudinal person-centered methodology to identify the dynamics of emergent profiles based on recovery-enhancing processes and their antecedents and outcomes across time. As such, our study helps to clarify the theoretical background of recovery-enhancing processes and their mutual relations.

## 2. Theoretical Background and Research Questions

There are several theories on the recovery process. The two most often referred models are the effort-recovery (E-R) model [1] and the conservation of resources (COR) theory [11]. The job demands–resources (JD-R) model [12] brings to these frameworks the roles of both job resources and demands in the context of recovery. Each framework contributes to the recovery research, and together, they give a more complete picture of how individuals recover from work strain.

According to the E-R model [1], time after work is important for the replenishment of psychological and physiological resources which have depleted due to job demands causing load reactions (e.g., fatigue). The essence of recovery is that when demands cease, load reactions are reduced, and the individual’s internal systems are able to return to baseline levels. If this is not the case, that is, if he or she is unable to replenish energies, an accumulated process may occur, resulting in physical and mental health problems in the long-term, e.g., [13]. When the E-R model primarily focuses on demands, the COR theory [11,14] focuses on resources that are personally valuable. If these resources are threatened or lost, stress-related outcomes are likely to follow. Recovery facilitates the replenishment of depleted resources (like energy). Therefore, time spent away from work engaging in various recovery activities and experiences have potential for enabling employees to restore the threatened or lost resources [15]. Those three recovery-enhancing processes examined in the present study function as replenishing depleted resources in line with the E-R model and the COR theory.

In the JD-R model [12,16,17], both job demands and job resources play an important role. First, job demands deplete the individual’s resources (like energy) and cause strain, which can lead to ill-being [16]. Second, resources provide and replenish energy enhancing well-being and help avoid the straining effects of job demands. Recovery is seen to function as an underlying mechanism in the relationship between resources or demands and relevant outcomes [18,19]. In the present study, both job demands and resources, as well as relevant well-being outcomes, are studied in line with the JD-R model.

Taken together, these three theories suggest that: (a) stopping work to recover from expended efforts due to job demands is critical for fostering well-being and performance, (b) recovery is a more active and motivational process than simply stopping work, and (c) job resources play an important role in mitigating the negative effects of job demands on employees’ outcomes. In addition, all theories suggest that the cycles of recovery are important in maintaining employee well-being. Thus, these theoretical frameworks complete each other and together provide a more detailed view of the recovery process for the study purposes.

### 2.1. Recovery-Enhancing Processes and Profiles

Recovery-enhancing processes that replenish resources during waking time can be divided into recovery activities and recovery experiences that may occur during different activities [20]. Recovery activities include hobbies or other activities people engage in during their leisure time (e.g., physical exercise), and recovery experiences are the processes of people’s experiences while they are engaged in those activities (e.g., psychological detachment from work while exercising). Earlier studies have shown that, of various free-time activities, physical activity is most recovering, e.g., [21,22,23]. Similarly, detachment from work during off-job time has turned out to be a core recovery experience; see [3,24] for reviews. Furthermore, sleep is fundamental for recovery [25] as during sleep fatigue is reduced and resources are restored [26]. Thus, although good sleep can be considered a recovery outcome, it is also a mechanism by which recovery occurs. In this study, we focus on these three—detachment from work, physical activity, and sleep—most promising processes for enhancing recovery based on the analysis offered by Sonnentag [3].

Psychological detachment from work means that an individual neither performs job-related tasks nor thinks about his or her job during nonwork time [20]. Over the years, psychological detachment from work has been found to be a particularly significant recovery experience; see [18,27,28] for meta-analyses. Psychological detachment from work has its roots in the E-R model because it implies that no further demands are made on the psycho-physiological systems that are called upon during work [20]. Disconnecting from work-related thoughts and stressors is crucial for recovery because, otherwise, the recovery process during off-job time may be hampered by prolonged psychophysiological activation [29].

Physical activity includes “all bodily movements that cause increases in physical exertion beyond that which occurs during normal activities or daily living” [30] (p. 4). Physical activity does not only have health benefits; see [31] for a review, but it also has beneficial effects on recovery, which has been shown quite consistently in previous studies, e.g., [21,22,23]. The benefits of physical exercise relate to both neurophysiological (e.g., endocrinological) and psychological processes (e.g., positive self-perceptions, psychological detachment from work [5,6,32,33]).

Sleep maintains well-being and is considered a vital period for psychological and physiological recovery processes [26]. Sleep quality can be defined as an overall subjective evaluation of one’s sleep on the basis of such criteria as an ease of falling asleep and sleeping well without frequent awakenings in the night [34]. Both sleep amount and quality are important for recovery. Nevertheless, sleep quantity does not necessarily mean that recovery outcomes are optimal; see [35] for a meta-analysis. Therefore, the quality of sleep is considered a better sign of “restorative” sleep, as good sleep reinvigorates psychological and physiological resources.

In the present study, we were interested to find out how recovery-enhancing processes operate together. Thus, we investigated how the three processes—detachment from work, physical activity, and sleep quality—combine within employees. For this purpose, we used latent profile analysis (LPA), of which the advantage is the forming of quantitatively and qualitatively distinct profiles [10,36]. Quantitatively distinct profiles vary in the level of the profile indicators (e.g., being high, moderate, or low on all constructs), whereas qualitatively distinct profiles vary in their shape (e.g., being high on one construct and low on other constructs). Thus, using LPA, we were able to identify quantitatively and/or qualitatively distinct subgroups of employees characterized by distinct combinations of recovery-enhancing processes [37]. Profile analysis is a valuable approach because it reflects the fact that people often pursue multiple activities in combination [38] and enjoy a mix of different recovery experiences [39,40]. In comparison to the variable-centered approach, the person-centered approach increases the specificity of the results as multiple subpopulations (i.e., profiles) are described separately, rather than the entire sample, as is done in the variable-centered approach. However, it is not the most parsimonious (simple to meaningfully interpret), as several parameters are produced [10,36]. Thus, the results are more precise, but less parsimonious than those produced by the variable-centered approach.

We do not pose formal hypotheses about specific profiles, given the explorative nature of the approach and the lack of earlier studies. Nevertheless, there are some profiles that seem theoretically plausible. First, based on the COR theory, some employees may be high in all processes, whereas some may be low in all processes. In line with the COR theory, all studied recovery processes can be seen as resources, which may form chains of resource gains or losses [11]. For example, physical activity promotes successful detachment from work during the evening and they both have a potential to enable good sleep. It may also be possible that good sleep and successful detachment are preconditions for high physical activity, as being physically active requires energetic resources. Thus, they form a chain of resource gains. Conversely, physical inactivity and poor detachment may reduce sleep quality, or poor sleep and poor detachment may appear as physical inactivity, and thereby constitute a chain of resource losses. Thus, we expect to find two quantitatively distinct profiles: high vs. low in all processes.

Second, some employees may be high in some and low in some other processes. There are several (altogether six) possibilities of how these qualitatively distinct profiles can emerge from the data. We expect to find such profiles in which good (or poor) sleep and successful (or poor) detachment exist together as in earlier meta-analyses [27,28], whereby their relationship turned out to be moderately high (r = 0.25–0.30). Nevertheless, other combinations (e.g., low physical activity and high detachment) and alone existing processes (e.g., high physical activity) are possible. In addition, as we have longitudinal data over two years, change (e.g., decreasing or increasing) profiles may also emerge. We assume that two years is not long enough to reveal fundamental changes in recovery-enhancing processes. Hence, we consider it likely that the data represent a mixture of different combinations of the three types of recovery processes, which tend to be stable across time.

Research Question 1: Which kind of distinct profiles of the recovery-enhancing processes (psychological detachment, physical activity, and sleep quality) exist that vary quantitatively (in level) and qualitatively (in shape)?

### 2.2. Job-Related Antecedents of Profiles

In the present study, we examined working hours, time pressure, and emotional demands, as demands require effort [16,17] and therefore deplete energy. Control and social support were examined as resources for reducing job demands and related strain, [16] and were therefore determined to have potential to replenish energy. These demands and resources are among the most important on the basis of the JD-R model [12] and they have shown to play a role in recovery processes.

Empirical and mostly cross-sectional evidence obtained in the 21st century suggests that job demands prevent, while job resources promote, the detachment process. The meta-analysis by Wendsche and Lohmann-Haislach [27] based on 91 samples in total revealed that job demands and detachment correlated significantly stronger than job resources and detachment. For job demands, the average correlation between detachment and quantitative demands (including time pressure) was highest (r = −0.28) and stronger than between detachment and working hours (r = −0.17). The average correlation between detachment and emotional demands (r = −0.22) fell in between these two. For job resources, detachment was, on average, more positively related to social support (r = 0.21) than to job control (r = 0.06). Longitudinal evidence is still rare, but at least two studies exit. First, in a one-year longitudinal study by Kinnunen and Feldt [41], high job demands (time pressure, decision-making demands, long working hours) predicted a decrease in psychological detachment over the years, but job resources (control, colleague support, justice of the supervisor) did not play a role. Second, Meier and Cho [42] showed that a high workload predicted a low level of psychological detachment after two months (controlling for the earlier detachment level). Thus, it seems that especially quantitative demands hamper detachment from work.

It has been shown that people who suffer from high job demands are likely to be less physically active during off-job time. Cross-sectional studies have found a negative relationship between working long hours and physical activity, e.g., [43,44]. This relationship might be explained by the fact that long working hours reduce time for off-job activities. Other studies using such measures of job demands as time pressure and emotional demands confirm these findings, e.g., [45,46]. In a meta-analysis by Fransson et al. [47], the risk for physical inactivity was higher in high-strain (i.e., high demands and low control) and passive (i.e., low demands and low control) jobs compared with low-strain (i.e., low demands and high control) and active (i.e., high demands and high control) jobs. In addition, people who were initially physically active became less physically active over time (across 2–9 years) when they worked in high-strain or passive jobs [47]. Thus, for staying physically active, job control tends to play a more important role than job demands. This may relate to the positive link between job control and feelings of self-determination that promote physical exercise [48]. As far as we know, the role of social support from colleagues and supervisors is unclear, but social support from friends regarding physical activity slightly enhances the future levels of physical activity; see [49] for a review. Theoretically, as social support has potential to replenish (not deplete) energy, supportive relations may save energy for physical activity.

Job demands and resources also relate to sleep quality. There are several reviews and meta-analyses, e.g., [35,50], showing that workload (including time pressure) and working hours link to sleep quality. In the meta-analysis by Litwiller and colleagues [35], consisting 152 mostly cross-sectional studies, high workload correlated significantly with poor sleep quality (r = −0.15) but the link to working hours (r = −0.05) was non-significant. Job control (r = 0.15) and social support (r = 0.06) showed positive associations with good sleep quality, with the correlation between job control and sleep quality being significant. Reviews on longitudinal research came to the conclusion that high job demands are linked to an increase in sleep problems over time [50,51]. Although there are also studies showing that poor sleep can have effects on perceived (higher) job demands and (lower) resources (control, social support) across time, e.g., [52,53], the majority of longitudinal studies suggest that high demands and low resources are antecedents for sleep problems.

On the basis of the JD-R model, we expect that profiles having the highest recovery-enhancing processes are associated with high levels of job resources and low levels of job demands, and profiles with the poorest recovery processes are associated with low job resources and high job demands. This occurs because job resources replenish resources (here: recovery-enhancing processes), whereas demands deplete them. As we cannot exactly predict the emerging profiles, we consider the following research question:

Research Question 2: Do job demands (working hours, time pressure, and emotional demands) and job resources (control, social support) predict the profiles of recovery-enhancing processes?

### 2.3. Well-Being Outcomes of Profiles

Of the recovery outcomes, we focused on job exhaustion, vigor at work, and vitality. They all relate to energy, which is a key resource to be depleted or replenished. Exhaustion and vigor are work-related energy constructs, whereas vitality—the subjective experience of being full of energy and alive [54]—is a context-free construct. Job exhaustion is described by the depletion of emotional resources and feelings of tiredness and fatigue resulting from work overload [55]. Additionally, poor recovery from work stress plays a role in maintaining exhaustion, which is considered the key dimension of job burnout [56]. Although exhaustion is the negative antithesis of vigor and feeling energized and enthusiastic about one’s job [57], it has been demonstrated that they are independent energy constructs [58,59]. Thus, feeling no exhaustion does not necessarily equate to feeling vigorous.

Psychological detachment from work during nonwork time correlates with various well-being indicators. Its correlations with indicators of poor well-being (i.e., exhaustion, fatigue) have turned out to be higher (between 0.36 and 0.42) than with vigor (r = 0.14) [18,27]. This suggests that psychological detachment might prevent better negative states than promote positive energetic states. The results of longitudinal studies are mixed. Some studies showed that poor psychological detachment predicted an increase in exhaustion over time lags between 6 and 12 months [60,61,62,63], but in other studies, poor psychological detachment did not predict change in exhaustion or fatigue over time lags between 4 and 24 months [41,64,65]. In addition, the relationship may be reversed as high exhaustion at baseline predicted a decrease in psychological detachment from work after 6 months [61]. Nevertheless, the long-term results suggest that psychological detachment may not predict vigor across one or two years or vice versa [41,64].

The positive effects of physical activity on physical and mental health are well documented; see [66] for a review. In addition, regular physical exercise is related to psychological (including measures of vigor and vitality) well-being—see [67] for a review—and subjective (including measures of positive affect) well-being—see [68] for a review. In a longitudinal study, high levels of physical exercise attenuated the association between increased burnout and increased depressive symptoms across several years [69]. Moreover, physically active employees have shown a lower level of burnout and felt more vigorous during working hours [70,71]. On the contrary, high levels of work-related exhaustion predicted a decrease in physical leisure activities over 12 months [72].

Sleep quality is also related to indicators of psychological and physical well-being. In a meta-analysis, correlations varied between 0.29 and 0.48 for poor sleep quality and various symptoms (e.g., anxiety, depression, fatigue), respectively [35]. A longitudinal research suggests that poor sleep plays a role in increasing job-related stress symptoms over time. On the one hand, insomnia predicted burnout symptoms 18 months later [73]. On the other hand, insomnia did not predict the onset of exhaustion symptoms across one year, but among employees who were initially exhausted, insomnia predicted the persistence of exhaustion symptoms across time [74]. This finding may suggest that poor sleep prevents people’s recovery process from job-related strain symptoms. Exhaustion may be a consequence of poor sleep, see [75] for a review, but exhaustion may also impair sleep. According to Sonnentag [3], exhaustion may impair sleep particularly through sleep hygiene behaviors (e.g., consuming coffee or alcohol, using electronic devices). Daily diary studies have indicated that, on days when employees slept better, they felt more vigorous or vital during the day [76,77,78].

On the basis of the COR theory [11], we assume that profiles with the highest recovery-enhancing processes are associated with the most positive outcomes, and, correspondingly, profiles with the poorest recovery-enhancing processes are associated with the most negative outcomes. Thus, employees with more recovery resources are better positioned for resource gains (here: vigor and vitality and less exhaustion), whereas employees with fewer recovery resources are more likely to experience resource losses (here: less vigor and vitality and more exhaustion). We present the following research question:

Research Question 3: Do profiles of recovery-enhancing processes differentially relate to employee well-being (job exhaustion, vigor at work, vitality) across time?

## 3. Method

### 3.1. Participants and Procedure

The study participants were employees working in 11 Finnish organizations from different sectors (e.g., education, information technology, and media). They were mainly recruited with the help of a company supplying occupational health services to organizations. The company selected those organizations which they thought to benefit from the project, i.e., in which recovery from work strain was an issue. In every organization, all employees were asked to participate. The questionnaire data were collected via electronic questionnaires once a year between 2013 and 2015. At Time 1, out of all the employees contacted (N = 3593), 1347 returned the completed questionnaire (response rate 37.5%). At Time 2, the completed questionnaire was returned by 841 (response rate 70.6%) of the employees who responded at Time 1 and who were still employed by the same organizations (N = 1192). At Time 3, 664 (response rate 83.1%) of the employees who responded at Times 1 and 2 and who were still employed by the same organizations responded (N = 799). Thus, the two-year longitudinal data with a one-year time lag consisted 664 participants. We used a one-year time lag between the measurements because it has been used commonly in earlier recovery studies, see [79] for a review, and because our aim was to study long-term processes. The employees were informed about the goals of the study and that their responses would be treated in confidence and that participation was voluntary.

The background factors of the longitudinal sample (N = 664) at Time 1 were as follows. Of the sample, 58% were women, the mean age was 47.5 years (range 23–66, SD = 9.90), and 38% held an academic degree (master’s level or higher). The majority (58%) worked in higher white-collar jobs (e.g., ICT specialists), 29.5% in lower-white collar jobs (e.g., nurses), 8.5% in blue-collar jobs (e.g., cleaners), and 4% as top managers. Most employees worked full-time (97%), were employed in a permanent job (91%), and worked on a regular day shift (90%). Working hours per week were on average 39 (range 12–60, SD = 5.90). Only 13% had a managerial position. Of the participants, 80% were living with a partner, and 44% had children (average of two) living at home.

### 3.2. Sample Attrition

The sample attrition was examined by comparing the background factors of the participants of the longitudinal sample with the non-respondents at Times 2 and 3. The participants and the non-respondents did not differ from each other in gender, education, occupational status, weekly working hours, managerial position, having a partner, or number of children. However, there were differences in the types of employment contract, working schedule, and age. The participants worked more often in a permanent employment contract (91% vs. 80%, *p* < 0.001), on a regular day shift (90% vs. 85%, *p* < 0.05), and were older (47.5 vs. 46.2 years, *p* < 0.05) than the non-respondents. The attrition analysis for most of the main study variables (psychological detachment, physical activity, sleep quality, time pressure, emotional job demands, job control, and vitality) revealed that the longitudinal sample did not differ from the non-respondents. However, the non-respondents reported lower levels of social support (3.91 vs. 4.00, *p* < 0.05), higher levels of exhaustion (2.06 vs. 1.86, *p* < 0.05), and lower levels of vigor (4.42 vs. 4.58, *p* < 0.05) than the participants.

### 3.3. Measures

Because this study was part of a larger project on recovery from work strain, the measures for each construct were kept short. Concerning each construct, we chose such items which have showed the highest loadings in earlier studies. Recovery-enhancing processes and well-being outcomes were measured at all three time points (T1–T3), and job-related antecedents were analyzed from T1.

Recovery-enhancing processes. Psychological detachment from work was evaluated with three items from the Recovery Experience Questionnaire [20], which has been validated in Finland both cross-sectionally [19] and longitudinally [80]. Participants responded to the items with respect to their off-job time (e.g., “I do not think about work at all”) using a 5-point scale (1 = totally disagree, 5 = totally agree). Physical activity was asked with one item: “How often do you spend free time on exercising at least 20 min with getting at least slightly out of breath and sweating?” The rating scale included six options: 1 = hardly ever or a few times per year, 2 = about once per month, 3 = a few times per month, 4 = about once per week, 5 = a few times per week, 6 = almost every day. Sleep quality was assessed with four items (“How often during the past month you have had (a) difficulties in falling asleep, (b) repeated awakenings, (c) too early awakenings and (d) feeling refreshed on waking up”) derived from the Karolinska Sleep Questionnaire and Sleep Quality Index [81,82] using a scale from 1 (very often or always) to 5 (very seldom or never).

Job-related antecedents. Of the job demands, we focused on working hours, time pressure, and emotional demands. Working hours were asked by one question: “How many hours do you actually work per week? (Include paid and unpaid overtime, but not your commuting time)”. Time pressure was evaluated three items (e.g., “How often does your job require you to work under time pressure?”) from the Quantitative Workload Inventory [83]. Emotional demands were also assessed with three items (e.g., “My work is emotionally demanding”) from the Copenhagen Psychosocial Questionnaire [84]. Of the job resources, control was measured with five items (e.g., “I can influence decisions that are important for my work”), and social support from colleagues and supervisors were measured with six items (e.g., “If needed, I can get support and help with my work from my coworkers/supervisor”) from the QPS Nordic-ADW [85]. All items of job-related antecedents were rated on a scale from 1 (very seldom or never) to 5 (very often or always).

Well-being outcomes. Job exhaustion was evaluated with the job exhaustion scale (5 items, e.g., “I feel emotionally drained from my work”) from the Maslach Burnout Inventory—General Survey [55], which has been validated in Finland [86]. Vigor at work was assessed by the shortened three-item Utrecht Work Engagement Scale (e.g., “At my work, I feel bursting with energy”) [57]. The response scale ranged from 0 (never) to 6 (every day) for both measures. Vitality was measured using a 4-item measure (e.g., “I feel alive and vital”) from the measure by Bostic et al. [87]. The items were rated on a 5-point scale (1 = very seldom, never; 5 = very often, always).

We paid attention to the following background factors: gender, age, occupational status, managerial position, living in a relationship and number of children living at home. These factors may be significant for recovery-enhancing processes [4] and/or well-being outcomes [88,89].

### 3.4. Statistical Analyses

The identification of longitudinal profiles based on three recovery-enhancing processes across two years was tested by LPA. The means and variances of the profile indicators, i.e., recovery-enhancing processes at T1–T3, were allowed to be freely estimated across the profiles as this led to a less biased and more proper presentation of the data according to the simulation studies [90,91]. Maximum likelihood with robust standard errors (MLR) [92]) was used as an estimation method for parameters of the profile solutions. Missing data were processed by using the full information maximum likelihood (FIML) method. All the analyses were conducted by using the Mplus 8.5 program [92].

The model fit was evaluated using various fit indices and tests: log likelihood, Akaike’s information criterion (AIC), the Bayesian information criterion (BIC), the adjusted Bayesian information criterion (aBIC), the Vuong–Lo–Mendell–Rubin test (VLMR), the Lo–Mendell–Rubin adjusted likelihood ratio test (LMR), the BLRT test (bootstrap likelihood ratio test), and entropy value [93,94]. The lower the AIC, BIC, and aBIC values, the better the model. The VLMR, LMR, and BLRT tests compare solutions with different numbers of profiles; a significant *p*-value (*p* < 0.05) indicates that the k–1 profile model is rejected in favor of the model with at least k profiles. The entropy (range 0–1) indicates the quality of the classification; values closer to 1.0 indicate a more reliable classification [95]. Besides these statistical criteria, we also considered the content, profile sizes, and theoretical interpretability of the different solutions as selection criteria [94].

To test predictors of recovery-enhancing processes, the auxiliary R3STEP method was used [96]. In this method, multinomial logistic regression is used to predict belonging to a profile with values of predictor variables (job demands and resources at T1). The interpretation is based on the beta coefficients and odds ratios which describe the change in likelihood of membership in a target profile versus a comparison profile [96,97]. To test the profile differences in the outcomes (job exhaustion, vigor at work, and vitality at T1–T3), auxiliary measurement-error-weighted-method (BCH) was used [98]. In the BCH, parameter comparison is based on the Wald chi-square test [98].

## 4. Results

### 4.1. Descriptive Results

Descriptive statistics, Cronbach’s alphas, and correlations of study variables at T1–T3 are in Table 1. Of the recovery-enhancing processes, detachment and sleep quality correlated positively with each other (r = 0.29–0.34, *p* < 0.001), but physical activity seemed to be separated from them, signaling that these processes are not redundant and they may form different combinations. Of the job demands, emotional demands had the highest correlations with detachment (r = −0.19–−0.24, *p* < 0.001) and sleep quality (r = −0.17–−0.21, *p* < 0.001). Moreover, of the job resources, both control (r = 0.21–0.29, *p* < 0.001) and social support (r = 0.26–0.29, *p* < 0.001) were associated the most with sleep quality. Sleep quality had the highest correlations with the outcomes, especially with job exhaustion (r = −0.50–−0.51, *p* < 0.001) and vitality (r = 0.46–0.50, *p* < 0.001), but also with vigor (r = 0.34–0.38, *p* < 0.001). Additionally, detachment (r = 0.24–0.28, *p* < 0.001) and physical activity (r = 0.19–0.22, *p* < 0.001) associated positively with vitality. Of the outcomes, vigor at work and vitality showed the highest mutual correlations (r = 0.63–0.66, *p* < 0.001).

### 4.2. Longitudinal Profiles of Recovery-Enhancing Processes

LPA was used to identify longitudinal profiles of recovery-enhancing processes. The fit indices and statistical tests from one-profile solution upward of seven are reported in Table 2. The eight-profile solution did not further converge, which is not unusual in complex models where variances between profiles are freely estimated. The VLMR and LMR tests supported the three-profile solution, but the BLRT did not support any specific profile solution. Moreover, the AIC, BIC, and aBIC values reached their minimum point for the seven-profile solution. However, their decrease became less pronounced after the three-profile solution. Entropy values for all estimated profile solutions were high. These results propose that the optimal number of profiles was between three and seven. Therefore, these five profile solutions were examined in more detail. Comparing the content of the profiles showed that for up to five profiles, each profile brought a meaningful addition to the model. However, the six- and seven-profile solutions produced no qualitatively new profiles, as the new profiles seemed to be arbitrarily divided into smaller and similarly shaped profiles of recovery-enhancing processes. Based on this inspection, the five-profile solution was determined to best describe the existing combinations of recovery-enhancing processes. The classification accuracy of this profile solution was high, that is, participants’ probability of being classified in their most likely profile varied between 90–93%.

The five-profile solution is graphically depicted in Figure 1. The first profile contained 15% of the participants and was characterized by a high detachment and average sleep quality, coupled with low physical activity at every measurement point. This profile was labelled ‘physically inactive, highly detaching’. The second profile was also characterized by a continued low physical activity, but compared with other profiles, employees in this profile also reported poor detachment and sleep quality. The profile, labelled ‘impaired recovery processes’, included 19% of the participants. The third profile had an opposite profile, as the participants belonging to this profile had both detachment and physical activity above an average level across all measurement points, and their sleep quality was the best compared with other profiles. Thus, the profile was labelled ‘enhanced recovery processes’ and it contained 25% of the participants. The fourth and smallest profile included 11% of the participants, who were characterized by poor detachment and very poor sleeping quality, but their physical activity was above an average across the two-year follow-up. This profile was labelled ‘physically active, poorly detaching and sleeping’. Finally, the fifth and the largest profile (29%) was characterized by average levels of detachment and sleep quality, coupled with above average physical activity. This profile was named ‘physically active’. To conclude, our results yielded altogether five different profiles of recovery-enhancing processes which were based both on different levels (two profiles) and shapes (three profiles), and they did not show change across time.

### 4.3. Antecedents of the Longitudinal Profiles of Recovery-Enhancing Processes

The results of a R3STEP auxiliary testing are presented in Table 3. In these analyses, the profiles of recovery-enhancing processes were predicted by background variables, job demands, and job resources (at T1). All predictors were estimated in the same model. The ‘enhanced recovery processes’ (3) profile was used as a reference profile in the analyses.

Looking first at the effects of background factors, none of them predicted the employee likelihood of membership into any of the profiles. With regard to the job-related predictors, we first noted that, of job demands, high emotional demands increased the participants’ likelihood to belong to the profiles of ‘impaired recovery processes’ (2) and ‘physically active, poorly detaching and sleeping’ (4) relative to the profile of ‘enhanced recovery processes’ (3). Of the job resources, higher job control perceptions predicted an increased likelihood of membership into the ‘enhanced recovery processes’ (3) profile relative to the profile of ‘physically active, poorly detaching and sleeping’ (4). Furthermore, higher levels of social support increased the likelihood of membership into the ‘enhanced recovery processes’ (3) profile relative to all other profiles. Finally, weekly working hours or time pressure presented no associations with their likelihood of membership into any of the profiles.

As an additional test, we also examined whether there were changes in job demands and resources over time. We found that only emotional demands changed (i.e., increased) over time. For this reason, the predictor role of emotional demands at T2 and T3 was also tested. After controlling the background variables, the participants with higher emotional demands at T2 or T3 had an increased likelihood of membership into the profiles of ‘impaired recovery processes’ (2) and ‘physically active, poorly detaching and sleeping’ (4) relative to the ‘enhanced recovery processes’ (3) profile. Thus, this result confirmed the findings using emotional demands at T1 as a predictor. Moreover, higher emotional demands at T2 or T3 increased the participants’ likelihood of membership into the ‘physically active’ (5) profile relative to the ‘enhanced recovery processes’ (3) profile.

To conclude, emotional demands, control, and social support played a role in predicting the likelihood of profile membership.

### 4.4. Differences in Well-Being Outcomes between the Longitudinal Profiles of Recovery-Enhancing Processes

Table 4 reports the differences between the profiles of recovery-enhancing processes in job exhaustion, vigor at work, and vitality at T1–T3. The Wald chi-square test demonstrated that there were statistically significant differences between the profiles in all well-being outcomes at every measurement time. Concerning all outcomes—job exhaustion, vigor at work, and vitality—the ‘enhanced recovery processes’ (3) profile had the highest well-being. They reported the lowest level of job exhaustion and the highest levels of vigor and vitality across time, and pairwise comparisons indicated that the profile differed significantly from all other profiles. The ‘physically active, poorly detaching and sleeping’ (4) profile, in turn, had the poorest well-being. They reported in every measurement the highest level of job exhaustion, which was at a significantly higher level than in all other profiles. In addition, their levels of vigor and vitality were lower than among those belonging to the ‘physically active’ (5) profile at every measurement point.

The ‘impaired recovery processes’ (2) profile had a lower level of job exhaustion than the ‘physically active, poorly detaching and sleeping’ (4) profile, but their levels of vigor and vitality did not differ significantly at T2 and T3. The ‘impaired recovery processes’ (2) profile also demonstrated higher levels of job exhaustion at T2 and T3 compared with the profiles of ‘physically inactive, highly detaching’ (1) and ‘physically active’ (5), and lower levels of vitality compared with the ‘physically active’ (5) profile. The profiles 1 and 5 did not differ from each other in job exhaustion, but the ‘physically active’ (5) profile had significantly higher levels of vigor at T1 and T3 and vitality at T2 compared with the ‘physically inactive, highly detaching’ (1) profile. To conclude, the results suggest that having more recovery-enhancing resources associates with a higher well-being.

## 5. Discussion

We utilized a person-centered approach [10,36] to identify profiles of recovery-enhancing processes (i.e., detachment from work, physical activity, and sleep). Utilizing this approach—and not a variable-centered approach—our aim was to extend the theoretical understanding of recovery-enhancing processes by showing that: (a) employees’ recovery-enhancing processes did not function in isolation, (b) job demands and resources had a predictive power in regard to the profile membership, (c) the profiles differed in well-being outcomes, and (d) they remained stable across two years. As such, our results contribute to the theoretical knowledge regarding recovery-enhancing processes in several ways.

First, we identified different combinations of recovery-enhancing processes over the follow-up time. The profiles that emerged were theoretically meaningful and provided new insights into the co-occurrence of recovery-enhancing processes. Two of the profiles (2 and 3) showed meaningful level differences, and three profiles (1, 4 and 5) differed in shape. In addition, all profiles were stable across time. This was expected as our study was conducted in stable circumstances (e.g., the participants continued their employment in the same organizations). Of the employees, 25% were classified as exemplifying the ideal profile, ‘enhanced recovery processes’, from the viewpoint of recovery theories [1,3,14]. Thus, the members of this profile had all recovery processes at a high level, suggesting that their recovery from work will very likely be successful. As the opposite, we also detected the ‘impaired recovery processes’ profile, consisting 19% of the participants, suggesting that a poor recovery from work will be likely among them. In line with the COR theory [11,14], these two quantitatively different profiles can be seen as examples of resource gain and resource loss chains: both resources and lack of resources accumulate.

In the rest of the profiles, it was typical that the profile members were high in some and average or low in some other recovery-enhancing processes. The smallest, ‘physically active, poorly detaching and sleeping’ profile, consisting 11% of the participants, showed the second worst combination of recovery processes, as they were poor in two processes (i.e., detachment and sleep), making them very likely vulnerable to poor recovery. The ‘physically inactive, highly detaching’ profile (15%), and the largest (29%), ‘physically active’ profile, were both high in one recovery process. Additionally, the members of the first profile were physically inactive and their sleep quality was at an average level, whereas in the latter profile, the two other recovery processes (i.e., detachment and sleep) were at an average level. Thus, as expected, there existed a mixture of different combinations of the three types of recovery processes.

Second, we examined how job demands and resources predicted profile membership. Theoretically, this was done by integrating job demands and resources from the JD-R model [16,17] to the recovery frameworks. We found that emotional demands, job control, and social support distinguished profile membership. Emotional demands made individuals more likely to belong to the profiles of ‘impaired recovery processes’, ‘physically active, poorly detaching and sleeping’, and ‘physically active’, whereas both higher social support and job control made individuals more likely to be in the ‘enhanced recovery processes’ profile. These results are in line with the JD-R model, according to which job demands deplete resources and job resources replenish them. The results also suggest that experiences at work do affect recovery choices at free time. Partially, the crucial role of emotional demands may relate to the characteristics of our sample, as we sought to study knowledge-intensive and emotionally demanding jobs in which recovery from work is known to be challenged [99,100]. The results also support the beneficial role played by both social support and control at work in recovery from work. Accordingly, they may promote physical activity [47], good sleep quality [35], and help detachment from work [27].

Third, we detected significant mean differences across profiles for the outcomes of job exhaustion, vigor at work, and vitality over time. The best results were attained when all three recovery processes functioned together, that is, the ‘enhanced recovery processes’ profile had the lowest job exhaustion and the highest vigor and vitality compared with all other profiles, as theoretically expected on the basis of the COR theory [14]. This result is also in line with earlier variable-centered studies, which have shown the beneficial effects of detachment [18,27], physical activity [66,67], and sleep [35] on well-being.

On the contrary, the worst well-being results did not prevail in the ‘impaired recovery processes’ profile, where all recovery-enhancing processes were low; however, in the ‘physically active, poorly detaching and sleeping’ profile, which reported more job exhaustion than all other profiles, their levels of vigor and vitality did not differ from the ‘impaired recovery processes’ profile. When examining the differences between these two profiles in detail, the striking difference is found in sleep quality. Although sleep quality is less than average in the ‘impaired recovery processes’ profile, it is very poor in the ‘physically active, poorly detaching and sleeping’ profile. Thus, although the profile members are physically active, poor sleep quality, in particular, exposes them to job exhaustion, referring to the crucial role of sleep on recovery from work [26].

It is also noteworthy that the ‘physically active’ profile—in which detachment and sleep quality were close to the average level of the whole sample but physical activity was high—had the second-best level of well-being. Although the profile’s well-being was lower than in the ‘enhanced recovery processes’ profile, the levels of vigor and vitality were reasonably high among the employees who belonged to it. Thus, average detachment and sleep quality, combined with physical activity, also produced a good result in well-being. This also occurred in the ‘physically inactive, highly detaching’ profile, in which high detachment was combined with low physical activity and average sleep. These two profiles did not differ from each other in job exhaustion, but the ‘physically active’ profile had significantly higher levels of vigor and vitality, suggesting that physical activity is more important in increasing vigor and vitality than reducing exhaustion. The result may also relate to the fact that energy is needed for being physically active. Thus, a combination in which at least one recovery-enhancing process is at a high level seems reasonably beneficial for well-being.

### 5.1. Strengths, Limitations and Future Directions

The strength of our study is that we used a person-centered approach to examine recovery-enhancing processes, whereas earlier studies have mainly investigated their separate associations or focused on profiles of recovery activities [38] or recovery experiences [39,40,80]. The main point of a person-centered research is to identify robust typical profiles and their meaningful connections to other constructs. Doing so, we learnt more about the formation of longitudinal profiles based on the three types of recovery-enhancing processes. Furthermore, we were able to show how these profiles were linked to theoretically potential antecedents (job demands and resources) and well-being outcomes. Nevertheless, in future studies, different recovery-enhancing processes (e.g., artistic activities, mastery experiences) could be studied. In addition, both the antecedents and outcomes examined could be broadened. For example, it would be interesting to study how organizational climate, such as psychosocial safety climate [101], is possibly associated with recovery processes and their combinations in certain organizations.

Our study was longitudinal, having three waves with a one-year time lag, which has most often been used in longitudinal recovery studies [79]. Nevertheless, it only covered two years, which may be too short to observe changes in recovery-enhancing processes. We also found no profiles with clear mean level changes. Therefore, we assume that in the future, longer time periods would be useful to capture potential long-term changes in recovery-enhancing processes. Having said that, it is also possible that more intensive measurements would be needed to better reveal the dynamics between the emerging profiles and their antecedents and outcomes.

Moreover, our study was based on self-reported data which expose common method variance. In our study, longitudinal design and the use of established scales may have reduced common method variance [102]. Nevertheless, in future research, the recommended objective measures to evaluate include physical activity or sleep. In fact, concerning each construct, a combination of subjective and objective measures would give a more complete overview on the construct and improve the validity of findings.

The response rates at T1–T3 were relatively low (from 18.5% to 37.5% relative to baseline respondents) jeopardizing the generalizability of our results. Dropout was related to background factors (temporary employment contract, other than day shift, and younger age) and to low social support, high exhaustion and low vigor, but dropout did not occur on the basis of recovery-enhancing processes (i.e., detachment, physical activity, and sleep quality). Thus, a healthy worker effect may play a role in our longitudinal sample. It is also noteworthy that our data were collected before the COVID-19 pandemic. Increased multi-locational work may have changed working habits, which may have affected recovery processes. Therefore, in future, the study should be replicated.

### 5.2. Practical Implications

Our research highlights the need for organizations and employees to promote detachment from work and good sleep in particular to maintain employee well-being. This can occur in several ways. First, organizations should keep job demands, especially emotional demands, at a reasonable level, and job resources should be kept at a sufficient level, as, in such working conditions, employees are more likely to detach from work after work and sleep well. Second, organizations may encourage employee recovery by such policies and norms that regulate working after working time and help keep work and leisure separated. Third, organizations could also provide, for example, gym memberships to facilitate exercise and training in sleep hygiene, as employees who make use of some of these resources will potentially be able to better recover from work and experience better sleep quality. Fourth, employees are recommended to apply diverse well-being-related self-leadership strategies to meet the demands of working life and enhance their well-being self-initially [103].

## 6. Conclusions

Employees do not use recovery-enhancing processes in isolation. Instead, the processes combine within people in rich ways. Our research advances recovery research by identifying five profiles of recovery-enhancing processes, detecting job-related demands and resources which can predict the profiles, and showing their well-being outcomes. The findings suggest that a profile in which at least one recovery-enhancing process is at a high level is reasonably beneficial for well-being, and that detachment from work and good sleep quality are more crucial recovery processes than physical activity.

## Figures and Tables

**Figure 1 ijerph-20-05382-f001:**
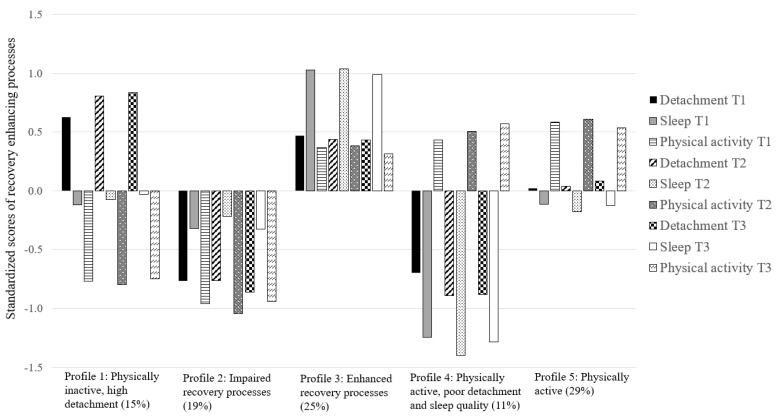
Longitudinal profiles of recovery-enhancing processes.

**Table 1 ijerph-20-05382-t001:** Means, Standard Deviations, Cronbach Alphas (on the Diagonal) and Correlations among the Study Variables.

Variables	M	SD	1	2	3	4	5	6	7	8	9	10	11	12	13	14	15	16	17	18	19	20	21	22	23
1 Working hours T1	38.99	5.90	-																						
2 Time pressure T1	3.87	0.82	0.28	0.88																					
3 Emotional demands T1	2.66	0.97	0.05	0.26	0.87																				
4 Control T1	3.21	0.82	0.06	−0.28	−0.14	0.78																			
5 Social support T1	4.00	0.65	0.03	−0.07	−0.19	0.32	0.80																		
6 Detachment T1	2.98	0.97	−0.24	−0.21	−0.24	0.02	0.16	0.86																	
7 Physical activity T1	4.54	1.22	−0.03	0.01	−0.07	0.08	0.05	0.13	-																
8 Sleep quality T1	3.45	0.92	−0.10	−0.22	−0.21	0.29	0.29	0.31	0.09	0.81															
9 Job exhaustion T1	1.86	1.43	0.17	0.38	0.29	−0.35	−0.34	−0.29	−0.09	−0.50	0.93														
10 Vigor T1	4.58	1.19	0.01	0.00	0.00	0.28	0.37	0.12	0.19	0.34	−0.45	0.89													
11 Vitality T1	3.41	0.78	0.05	−0.10	−0.10	0.27	0.37	0.28	0.22	0.48	−0.52	0.63	0.88												
12 Detachment T2	3.03	0.97	−0.18	−0.18	−0.21	0.05	0.14	0.62	0.02	0.27	−0.20	0.09	0.19	0.86											
13 Physical activity T2	4.50	1.25	0.01	0.02	−0.07	0.06	0.03	0.08	0.70	0.07	−0.05	0.10	0.16	0.04	-										
14 Sleep quality T2	3.40	0.93	−0.05	−0.16	−0.18	0.22	0.26	0.23	0.03	0.73	−0.44	0.29	0.41	.029	0.01	0.81									
15 Job exhaustion T2	1.90	1.42	0.13	0.33	0.23	−0.30	−0.28	−0.24	−0.08	−0.43	0.69	−0.35	−0.44	−0.25	−0.06	−0.50	0.93								
16 Vigor T2	4.44	1.30	0.04	0.05	0.01	0.18	0.29	0.12	0.18	0.34	−0.34	0.68	0.57	0.14	0.10	0.38	−0.45	0.91							
17 Vitality T2	3.45	0.80	−0.02	−0.09	−0.12	0.24	0.28	0.20	0.21	0.41	−0.42	0.51	0.69	0.24	0.19	0.50	−0.58	0.65	0.88						
18 Detachment T3	3.01	0.95	−0.12	−0.12	−0.19	0.00	0.14	0.59	0.03	0.24	−0.17	0.08	0.14	0.65	0.06	0.27	−0.19	0.12	0.18	0.85					
19 Physical activity T3	4.56	1.20	0.03	0.01	0.01	0.06	0.06	0.05	0.61	−0.04	−0.05	0.14	0.14	0.01	0.67	0.01	−0.07	0.11	0.13	0.03	-				
20 Sleep quality T3	3.38	0.91	−0.05	−0.13	−0.17	0.21	0.26	−0.22	0.07	0.69	−0.36	0.27	0.38	0.26	0.03	0.73	−0.40	0.32	0.40	0.34	0.02	0.79			
21 Job exhaustion T3	1.92	1.44	0.09	0.26	0.24	−0.23	−0.24	−0.19	−0.09	−0.39	0.58	−0.31	−0.37	−0.21	−0.07	−0.46	0.70	−0.37	−0.47	−0.28	−0.08	−0.51	0.93		
22 Vigor T3	4.44	1.32	0.01	0.06	0.02	0.18	0.30	0.11	0.15	0.27	−0.28	0.63	0.49	0.11	0.09	0.33	−0.37	0.71	0.54	0.17	0.13	0.35	−0.49	0.91	
23 Vitality T3	3.39	0.81	0.01	−0.04	−0.05	0.20	0.29	0.22	0.23	0.34	−0.36	0.49	0.62	0.20	0.15	0.38	−0.47	0.54	0.69	0.28	0.20	0.46	−0.59	0.66	0.90

Note. If r ≥ |0.13|, *p* < 0.001.

**Table 2 ijerph-20-05382-t002:** Enumeration of Fit Statistics for Longitudinal Profiles of Recovery-Enhancing Processes.

Nr. Of Profiles	LL	FP	AIC	BIC	aBIC	VLMR (*p*)	LMR (*p*)	BLRT (*p*)	Entropy	Latent Profile Proportions %
1	−8556.33	18	17,148.66	17,229.63	17,172.48	-	-	-	-	100
2	−8025.81	37	16,125.63	16,292.07	16,174.59	<0.001	<0.001	<0.001	0.83	38/62
3	−7677.86	56	15,467.72	15,719.62	15,541.82	<0.01	<0.01	<0.001	0.85	31/33/36
4	−7517.09	75	15,184.18	15,521.55	15,283.42	0.256	0.259	<0.001	0.85	25/33/26/16
5	−7399.99	94	14,987.98	15,410.82	15,112.37	0.191	0.192	<0.001	0.85	15/19/25/11/29
6	−7291.82	113	14,809.65	15,317.96	14,959.18	0.191	0.192	<0.001	0.85	12/15/21/21/19/12
7	−7207.06	132	14,678.12	15,271.89	14,852.78	0.520	0.521	<0.001	0.86	9/13/11/17/16/23/11

Note. LL = log-likelihood; FP = free parameters; AIC = Akaike information criterion; BIC = Bayesian information criterion; aBIC = sample-size adjusted Bayesian information criterion; VLMR = Vuong-Lo-Mendell-Rubin test; LMR = Lo-Mendell-Rubin test; BLRT = Bootstrapped likelihood ratio test.

**Table 3 ijerph-20-05382-t003:** Predictors of Longitudinal Profiles of Recovery-Enhancing Processes: Three-Step Results (R3STEP).

Predictors	Profile 1 vs. Profile 3				Profile 2 vs. Profile 3				Profile 4 vs. Profile 3				Profile 5 vs. Profile 3			
	Coef.	S.E.	*p* Value	OR	Coef.	S.E.	*p* Value	OR	Coef.	S.E.	*p* Value	OR	Coef.	S.E.	*p* Value	OR
Gender ^1^	−0.41	0.35	0.242	0.66	0.08	0.29	0.780	1.08	−0.45	0.39	0.246	0.64	−0.24	0.28	0.399	0.79
Age in years	0.02	0.02	0.255	1.02	0.01	0.02	0.640	1.01	0.00	0.02	0.940	1.00	−0.02	0.01	0.084	0.98
Relationship status ^2^	−0.69	0.37	0.063	0.50	0.59	0.42	0.166	1.80	−0.71	0.41	0.083	0.49	−0.09	0.35	0.805	0.91
Number of children ^3^	0.08	0.16	0.629	1.08	0.02	0.13	0.891	1.02	−0.05	0.18	0.775	0.95	0.06	0.14	0.662	1.06
Occupational status ^4^	−0.23	0.26	0.384	0.79	−0.09	0.25	0.717	0.91	0.23	0.28	0.410	1.26	−0.23	0.22	0.306	0.79
Managerial position ^5^	−0.54	0.64	0.398	0.58	−0.29	0.48	0.539	0.75	−0.20	0.59	0.729	0.82	−0.09	0.47	0.848	0.91
Working hours per week	−0.01	0.02	0.831	0.99	0.05	0.03	0.133	1.05	0.03	0.03	0.315	1.03	0.02	0.03	0.543	1.02
Time pressure	0.01	0.24	0.974	1.01	0.18	0.19	0.342	1.20	0.41	0.25	0.105	1.51	0.33	0.18	0.068	1.39
Emotional demands	0.22	0.21	0.303	1.25	0.37	0.15	0.017	1.45	0.47	0.22	0.032	1.60	0.28	0.15	0.061	1.32
Control	−0.33	0.24	0.177	0.72	−0.28	0.22	0.207	0.76	−0.56	0.23	0.013	0.57	−0.21	0.19	0.272	0.81
Social support	−0.84	0.26	0.001	0.43	−0.61	0.27	0.022	0.54	−1.32	0.31	0.001	0.27	−0.49	0.24	0.043	0.61

Note. The Well-functioning recovery processes (3) profile was used as a reference profile. ^1^ Gender (1 = woman, 2 = man), ^2^ relationship status (1 = no relationship, 2 = in a relationship), ^3^ number of children living at home, ^4^ occupational status (1 = blue-collar worker, 2 = lower white-collar worker, 3 = higher white-collar worker, 4 = top manager), ^5^ managerial position (1 = no, 2 = yes). Coef. = the estimate (β) from the R3STEP multinomial logistic regression analysis: Positive estimate values describe a higher likelihood of belonging to the first latent profile, whereas negative estimate values indicate a higher likelihood of belonging to the second latent profile of the two comparison profiles. S.E. = standard error, OR = odds ratio. Statistically significant differences are bolded.

**Table 4 ijerph-20-05382-t004:** Differences in Longitudinal Profiles of Recovery-Enhancing Processes in Job-Related Exhaustion and Vigor, and Vitality across Time.

Distal Outcome Variable	Profile 1 *M* (S.E)	Profile 2 *M* (S.E)	Profile 3 *M* (S.E)	Profile 4 *M* (S.E)	Profile 5 *M* (S.E)	Wald χ^2^/ *p*-Value	Profile Differences
Exhaustion T1	1.89 (0.16)	2.18 (0.14)	1.01 (0.09)	2.95 (0.20)	1.93 (0.10)	115.71, *p* < 0.001	3 < 1 ***, 2 ***, 4 ***, 5 *** 4 > 1 ***, 2 **, 5 ***
Vigor T1	4.21 (0.15)	3.96 (0.14)	5.17 (0.07)	3.89 (0.18)	4.47 (0.09)	103.51, *p* < 0.001	3 > 1 ***, 2 ***, 4 ***, 5 *** 5 > 1 **, 4 **
Vitality T1	3.21 (0.08)	3.16 (0.08)	3.96 (0.05)	2.98 (0.09)	3.39 (0.06)	134.71, *p* < 0.001	3 > 1 ***, 2 ***, 4 ***, 5 *** 5 > 2 *, 4 ***
Exhaustion T2	1.84 (0.15)	2.39 (0.15)	0.97 (0.08)	2.97 (0.19)	1.98 (0.10)	155.95, *p* < 0.001	3 < 1 ***, 2 ***, 4 ***, 5 *** 4 > 1 ***, 2 *, 5 * 2 > 1 *, 5 *
Vigor T2	4.21 (0.15)	3.96 (0.14)	5.17 (0.07)	3.89 (0.18)	4.47 (0.09)	103.51, *p* < 0.001	3 > 1 ***, 2 ***, 4 ***, 5 *** 5 > 2 **, 4 **
Vitality T2	3.22 (0.09)	3.09 (0.08)	4.05 (0.05)	3.02 (0.08)	3.44 (0.06)	173.74, *p* < 0.001	3 > 1 ***, 2 ***, 4 ***, 5 *** 5 > 1 *, 2 ***, 4 ***
Exhaustion T3	1.80 (0.14)	2.41 (0.16)	1.08 (0.09)	3.21 (0.21)	1.86 (0.10)	130.14, *p* < 0.001	3 < 1 ***, 2 ***, 4 ***, 5 *** 4 > 1 ***, 2 **, 5 *** 2 > 1 **, 5 **
Vigor T3	4.20 (0.14)	4.06 (0.14)	5.06 (0.08)	3.66 (0.21)	4.58 (0.09)	75.39, *p* < 0.001	3 > 1 ***, 2 ***, 4 ***, 5 *** 5 > 1 *, 2 **, 4 ***
Vitality T3	3.27 (0.08)	3.04 (0.08)	3.88 (0.06)	2.88 (0.10)	3.45 (0.06)	113.01, *p* < 0.001	3 > 1 ***, 2 ***, 4 ***, 5 *** 5 > 2 ***, 4 *** 1 > 2 *, 4 **

Note. Analyses were run utilizing the BCH procedure in Mplus. Response scale 0–6 for exhaustion and vigor, and 1–5 for vitality. * *p* < 0.05; ** *p* < 0.01; *** *p* < 0.001.

## Data Availability

The data is available from Ulla Kinnunen upon reasonable request.

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
