# Peer review of "Longitudinal Profiles of Recovery-Enhancing Processes: Job-Related Antecedents and Well-Being Outcomes"

_ijerph, 2023, doi:10.3390/ijerph20075382_

Round 1

Reviewer 1 Report

1.  The authors stated the purpose of their research as follows:

The aim of this study is to examine recovery profiles based on psychological detachment from work, physical exercise, and sleep. In addition to identifying longitudinal profiles (across two years) of psychological detachment from work, physical exercise, and sleep, the authors investigate antecedent factors (job demands and resources) and well-being outcomes (job exhaustion, vigor at work, and vitality).

This is a very important problem in practice and theory.

2. The authors did not clearly formulate the research problem. Please emphasize in the introduction why your research is important. What factors of the work environment affect the employee? Why the employee is overworked. Please pay attention to stressful work conditions, increasing demands, digitization ect. This will better outline the background of the research undertaken.

3. The subject of research is original and well defined. The results represent progress in current knowledge.

4. The discussion refers to other research and has references to the theory that the author wants to develop.

5. The literature review is appropriate, but please include more recent publications.

6. The manuscript is clear, relevant for the field, scientifically sound and presented in a well-structured manner. The figures and tables are appropriate.

7. In conclusion, the article should be slightly improved.

Author Response

Review 1

Comment 1: “The authors stated the purpose of their research as follows: The aim of this study is to examine recovery profiles based on psychological detachment from work, physical exercise, and sleep. In addition to identifying longitudinal profiles (across two years) of psychological detachment from work, physical exercise, and sleep, the authors investigate antecedent factors (job demands and resources) and well-being outcomes (job exhaustion, vigor at work, and vitality). This is a very important problem in practice and theory.”

Response: We thank you for your positive and helpful comments and for taking the time to point out ways to improve our manuscript – your response and support are highly appreciated.

Comment 2. “The authors did not clearly formulate the research problem. Please emphasize in the introduction why your research is important. What factors of the work environment affect the employee? Why the employee is overworked. Please pay attention to stressful work conditions, increasing demands, digitization ect. This will better outline the background of the research undertaken.”

Response: Thank you for this important comment. As suggested, we have added information about the demands of current working life on page 1 to show the importance of recovery – the main target of our study.

Comment 3. “The subject of research is original and well defined. The results represent progress in current knowledge.”

Response: We are grateful to the reviewer for the encouraging feedback.

Comment 4. “The discussion refers to other research and has references to the theory that the author wants to develop.”

Response: Thank you for pointing out the strengths of our discussion.

Comment 5. “The literature review is appropriate, but please include more recent publications.”

Response: Thank you for this comment that we seriously pondered. However, in our opinion, the references cover the most important ones from the point of view of the topic. Moreover, our paper has already quite many references in its current form and includes also the recent ones (from years 2020-22). Therefore, we did not add new references. However, if it is deemed necessary, we are willing to reconsider this issue. 

Comment 6. “The manuscript is clear, relevant for the field, scientifically sound and presented in a well-structured manner. The figures and tables are appropriate.”

Response: Thank you again for this positive evaluation.

Comment 7. “In conclusion, the article should be slightly improved.”

Response: We wish to thank you once again for your thoughtful comments. We appreciate your support for our work, and hope that you approve of the changes we have made.

Reviewer 2 Report

Dear authors,

The study is well-structured and written and has a great potential to provide significant contribution to literature on Recovery-Enhancing processes. However, some few observations and concerns need to be addressed.

Introduction

The introduction of a study sets the foundation upon which the entire study is built, and I think the researchers need to address the following in the introduction section;

First, I think they need to provide more valid reasons for adopting the person centered approach instead of the variable centered approach that is very common in literature. This could be addressed by further highlighting some weaknesses that the variable centered approach may have and providing more benefits that the person centered approach could offer to research.

The authors could highlight more on the contributions of the study. First, they could briefly provide an explanation to their stated contributions and second, I feel like the study has several added benefits to both theory and practice that the authors could include. For eg, longitudinal studies

Theoretical background

Besides that the E-R and the COR theory are often the most used theories, why did the authors focused on these two theories? Also, in addition to the JD-R model, how does these three theories help us to specifically understand recovering processes. I think the authors need to provide a bit more justification to this.

There came up a question:

I was wondering about the order of the diverse combined recover processes: ? Should some profiles maybe first restore energy to be able to exercise physically? Might there be a comprehensive order. Can this be taken more into account theoretically and empirically? Which would be for the diverse profiles the best order? Could this be more included in the study?

Methodology

Regarding the recruitment for participants for the study, was it done by randomly selecting participants? or they were conveniently selected or otherwise? I think providing information on the sampling technique may be more useful here.

Result and discussion

Results were clear and the findings were comprehensively discussed. Thus, no further comments on the results and discussion.

Practical implications and conclusion

No further comment here

Footnote: 

I think that the footnote could be added as additional findings and included as the last paragraph of the discussion section. I think the findings are very important and may contribute significantly to literature on recovery enhancing processes.

Grammatical errors 

The paper was well written with no or few errors with the exception of this. 

On page 13, the third paragraph, two commas were used instead of one (poor detaching and sleeping,,)

Besides this, there were no further errors identified. 

Author Response

Review 2

“Dear authors,

The study is well-structured and written and has a great potential to provide significant contribution to literature on Recovery-Enhancing processes. However, some few observations and concerns need to be addressed.”

Response: We thank you for your positive and encouraging feedback, as well as helpful comments which are highly appreciated.

“Introduction

The introduction of a study sets the foundation upon which the entire study is built, and I think the researchers need to address the following in the introduction section;

First, I think they need to provide more valid reasons for adopting the person centered approach instead of the variable centered approach that is very common in literature. This could be addressed by further highlighting some weaknesses that the variable centered approach may have and providing more benefits that the person centered approach could offer to research.

The authors could highlight more on the contributions of the study. First, they could briefly provide an explanation to their stated contributions and second, I feel like the study has several added benefits to both theory and practice that the authors could include. For eg, longitudinal studies”

Response: Thank you for these valuable comments. Based on them, we have added reasons for adopting the person-centered approach on page 2. Moreover, on page 4 we make some comparisons between variable- and person-centered approaches and show some of their pros and cons. Furthermore, we have strengthened the contributions of our study on page 2.

“Theoretical background

Besides that the E-R and the COR theory are often the most used theories, why did the authors focused on these two theories? Also, in addition to the JD-R model, how does these three theories help us to specifically understand recovering processes. I think the authors need to provide a bit more justification to this.

There came up a question:

I was wondering about the order of the diverse combined recover processes: ? Should some profiles maybe first restore energy to be able to exercise physically? Might there be a comprehensive order. Can this be taken more into account theoretically and empirically? Which would be for the diverse profiles the best order? Could this be more included in the study?”

Response: Thank you for encouraging us to further articulate the theoretical background of our study. In line with the suggestion, we have now added explanation for the choice of the theories by summarizing their main points which are crucial for our study (page 3). The question of the order of the diverse combined recovery processes is important. Unfortunately, we cannot answer the question with our data and the current methodology used. We have, however, added the idea that energy is needed for physical activity on page 4 where we present possibly emerging profiles. We also refer to this interpretation in the discussion (page 15). Also, in the discussion (page 15) we suggest that more intensive studies would be needed to reveal better the dynamics between the profiles.

“Methodology

Regarding the recruitment for participants for the study, was it done by randomly selecting participants? or they were conveniently selected or otherwise? I think providing information on the sampling technique may be more useful here.”

Response: Thank you for this important comment. We have now added text about the recruitment of the participants on page 7. As the company supplying occupational health services to organizations helped us to find the participating organizations among their clients, the recruitment did not happen randomly.

“Result and discussion

Results were clear and the findings were comprehensively discussed. Thus, no further comments on the results and discussion.”

Response: Thank you for your positive feedback.

“Practical implications and conclusion

No further comment here”

Response: Thank you for considering our implications and conclusions as sufficient.

“Footnote: 

I think that the footnote could be added as additional findings and included as the last paragraph of the discussion section. I think the findings are very important and may contribute significantly to literature on recovery enhancing processes.”

Response: As suggested, the text in the footnote has been integrated to the text (page 12).

“Grammatical errors 

The paper was well written with no or few errors with the exception of this. On page 13, the third paragraph, two commas were used instead of one (poor detaching and sleeping,,). Besides this, there were no further errors identified. 

Response: The grammatical error on page 13 has been corrected.